# Beryllium Radioactive Isotopes as a Probe to Measure the Residence Time of Cosmic Rays in the Galaxy and Halo Thickness: A "Data-Driven" Approach [†]

**Francesco Nozzoli** [1,*] and **Cinzia Cernetti** [2]

1 Istituto Nazionale Fisica Nucleare, INFN-TIFPA, via Sommarive, 14, I-38123 Trento, Italy
2 Department of Physics, Trento University, via Sommarive, 14, I-38123 Trento, Italy; cinzia91.cernetti@gmail.com
* Correspondence: Francesco.Nozzoli@unitn.it
† This paper is an extended version from the proceeding paper: Cinzia Cernetti and Francesco Nozzoli. Beryllium Radioactive Isotopes as a Probe to Measure Residence Time of Cosmic Rays in the Galaxy and Halo Thickness. In Proceedings of the 1st Electronic Conference on Universe, online, 22–28 February 2021.

**Abstract:** Cosmic rays are a powerful tool for the investigation of the structure of the magnetic fields in the Galactic halo and the properties of the inter-stellar medium. Two parameters of the cosmic ray propagation models, the Galactic halo (half) thickness, $H$, and the diffusion coefficient, $D$, are loosely constrained by current cosmic ray flux measurements; in particular, a large degeneracy exists, with only $H/D$ being well measured. The $^{10}Be/^{9}Be$ isotopic flux ratio (thanks to the 2 My lifetime of $^{10}Be$) can be used as a radioactive clock providing the measurement of cosmic ray residence time in a galaxy. This is an important probe with which to solve the $H/D$ degeneracy. Past measurements of $^{10}Be/^{9}Be$ isotopic flux ratios in cosmic rays are scarce, and were limited to low energy and affected by large uncertainties. Here a new technique to measure $^{10}Be/^{9}Be$ isotopic flux ratio, with a data-driven approach in magnetic spectrometers is presented. As an example, by applying the method to beryllium events published via PAMELA experiment, it is now possible to determine the important $^{10}Be/^{9}Be$ measurement while avoiding the prohibitive uncertainties coming from Monte Carlo simulations. It is shown how the accuracy of PAMELA data strengthens the experimental indication for the relativistic time dilation of $^{10}Be$ decay in cosmic rays; this should improve the knowledge of the $H$ parameter.

**Keywords:** astroparticle physics; cosmic rays; galactic halo

## 1. Introduction

Cosmic rays are a powerful tool for investigations of physics/astrophysics: high-energy cosmic ray composition provides information on the galactic PeVatrons, and the small anti-matter component in a cosmic ray could identify the dark matter annihilation in our galaxy.

The structure of the magnetic fields in the galactic halo and the properties of the inter-stellar medium can be probed by detailed cosmic ray flux measurements. In particular, the ratio of secondary cosmic rays (such as Li, Be and B) over the primary cosmic rays (such as He, C and O) allows one to determine the grammage —that is, the amount of material passed through by cosmic rays on their journeys through the Galaxy.

Two parameters of the cosmic ray propagation models, the galactic halo (half) thickness, $H$, and the diffusion coefficient, $D$, are loosely constrained by the grammage measurement; in particular, a large degeneracy exists, with only $H/D$ being well measured [1].

The uncertainties of $D$ and $H$ parameters (the latter is known to be in the 3–8 kpc range) also reflect on the accuracy of the determination of secondary anti-proton and positrons fluxes that are the background for the dark matter or exotic (astro-)physics searches [2–5].

Abundances of long-living unstable isotopes in cosmic rays can be used as radioactive clocks providing measurements of cosmic rays' time spent in the Milky Way. This time information is complementary to the crossed grammage; thus, the abundance of radioactive isotopes in cosmic rays is an important probe for solving the existing $H/D$ degeneracy in cosmic ray propagation models.

*Beryllium Isotopic Measurements in Cosmic Rays*

Only a few elements in cosmic rays, namely, Be, Al, Cl, Mg and Fe, contain long-living radioactive isotopes; among them, beryllium is the lightest, and thus the most promising for measurements of isotopic composition in the relativistic kinetic energy range. Three beryllium isotopes are found in cosmic rays:

- $^7$Be: stable bare nucleus. It decays by electron capture ($T_{1/2}$ = 53 days).
- $^9$Be: stable.
- $^{10}$Be $\beta$-radioactive nucleus ($T_{1/2}$ = 1.39$\times$10$^6$ years).

The missing $^8$Be has a central role in stellar and Big-Bang nucleosynthesis; the extremely short half-life ($8.19 \times 10^{-17}$ s) represents a bottleneck for the efficient synthesis of heavier nuclei in the Universe. From the measurement point of view, this "isotopic hole" in the beryllium mass spectrum is very useful in order to determine the large amount of $^7$Be and reduce the contamination in the identification of $^9$Be and $^{10}$Be.

It is important to mention that with $^{10}$B being the daughter nucleus of $^{10}$Be $\beta$-decay, it is possible to extract information of the parameters $H$ and $D$ by using the precise measurement of the elemental ratio Be/B in place of the much more difficult measurement of the $^{10}$Be/$^9$Be isotopic ratio. This possibility was originally discussed in [6], and more recently, the expected value of the $^{10}$Be/$^9$Be isotopic ratio in cosmic rays has become able to be inferred from the precisely measured cosmic rays elemental ratios [1,2,4,5]. However, the problem of uncertainties or biases in fragmentation cross-sections should also be mentioned; in particular, nuclear uncertainties in secondary production models are a major limitation in the interpretation of secondary cosmic ray nuclei [5,7,8]. Thus, isotopically resolved cosmic ray measurements such as the $^{10}$Be/$^9$Be ratio remain very important to model the propagation of cosmic rays.

An example of a magnetic spectrometer able to measure cosmic ray isotopic composition in space is given by the PAMELA detector (see Figure 1). The PAMELA spectrometer was installed onboard Resurs-DK1, the Russian satellite, and launched on 15 June 2006. The spectrometer is 1.3 m long; the 0.43T permanent magnet is equipped with a silicon microstrip Tracker that provides particle rigidity ($R = p/Z$) up to 1TV and energy loss (dE/dx) measurements. A time of flight (ToF) detector, made of three pairs of plastic scintillators, is used to measure the velocity, $\beta = v/c$; particle energy loss, dE/dx; and charge, Z. The time resolution of ToF ranges from 85 to 80 ps for lithium and beryllium nuclei, respectively. At its bottom a silicon-tungsten imaging calorimeter is able to provide a redundant $\beta$ measurement for sub-relativistic particles thanks to the Bethe–Bloch formula dE/dx $\sim Z^2/\beta^2$. The identification of different isotopes in magnetic spectrometers relies on the simultaneous measurement of particle rigidity and velocity. This allows the reconstruction of the particle mass $m = RZ/(\gamma\beta)$.

The typical mass resolution of magnetic spectrometers onboard past or current cosmic ray experiments ($\delta M \simeq 0.4$–1 amu) does not allow for an event-by-event isotope identification; therefore, the "traditional" approach for the measurement of beryllium's isotopic abundances relies on the reading of the experimental mass distribution with a Monte Carlo simulation.

This approach requires a very well tuned Monte Carlo simulation of the experiment, and the possible small residual discrepancies with the real detector response could prevent the measurement of the (interesting) small amount of $^{10}$Be.

The analysis of lithium and beryllium isotopes collected by the PAMELA experiment between July 2006 and September 2014 is well described in [9], where the "Monte-Carlo-

based" approach only allows for the measurement of $^7Li/^6Li$ and $^7Be/(^9Be+^{10}Be)$ due to a slight mismatch of Monte Carlo distributions with the flight data.

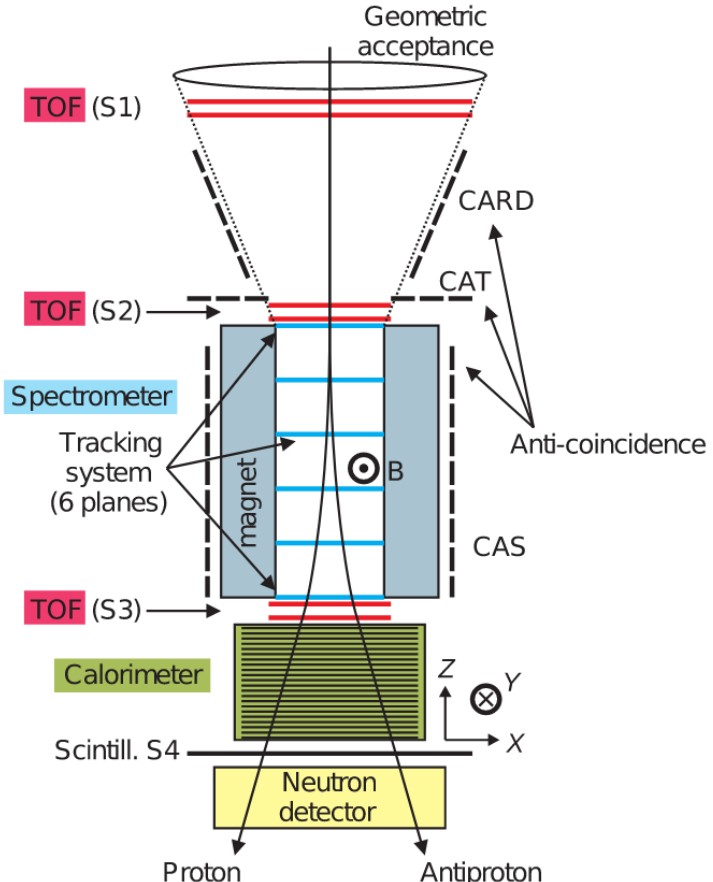

**Figure 1.** Scheme of the PAMELA detector. The spectrometer is 1.3 m high; the 0.43T permanent magnet is equipped with a silicon microstrip Tracker that provides rigidity up to 1TV and dE/dx information. A time of flight (ToF) detector, made of three pairs of plastic scintillators, can measure the velocity and charge of the particle. At its bottom, a silicon-tungsten imaging Calorimeter is able to provide a redundant $\beta$ evaluation, for sub-relativistic particles, via dE/dx measurement.

In the following a new data-driven approach for the measurement of beryllium isotopic abundances with magnetic spectrometers is described; this can evade part of the systematics related to Monte Carlo simulation. As an example, the application of this approach to PAMELA lithium and beryllium event counts, gathered from Figures 3 and 4 of [9], is shown; and a preliminary, new measurement of $^{10}Be/^9Be$ in the 0.2–0.85 GeV/n range is provided. Figure 3 of [9] contains the numbers of lithium and beryllium events as functions of $\beta$ measured by the time of flight (ToF) sub-detector and rigidity measured by the silicon tracker, whereas Figure 4 of [9] contains the numbers of lithium and beryllium events as functions of dE/dx measured by the imaging calorimeter and rigidity measured by the silicon tracker. Both figures contain color coded small integers; therefore, for this example, counts could be extracted (without errors) by image analysis as if they were published in a table.

## 2. *Data-Driven* **Analysis**

By knowing the true values of beryllium isotope masses and having a physically motivated scaling of the mass resolution for the three beryllium isotopes, the shapes of the isotope mass distributions can be retrieved, by self-consistency, solely from the measured data.

In particular, the expected mass resolution for a magnetic spectrometer is:

$$\frac{\delta M}{M} = \sqrt{\left(\frac{\delta R}{R}\right)^2 + \gamma^4 \left(\frac{\delta \beta}{\beta}\right)^2}. \tag{1}$$

Typically, the isotopic measurement is pursued in kinetic energy/nucleon bins (i.e., in $\beta$ bins); therefore, the velocity contribution to mass resolution is constant for the different isotopes.

Moreover, in the (low) kinetic energy range accessible by current isotopic measurements, the rigidity resolution is dominated by multiple Coulomb scattering; i.e., $\delta R/R$ is practically constant for the different isotopes.

Finally, the masses of the three Be isotopes are within 30%; therefore, for a fixed $\beta$ value, the rigidity values for different Be isotopes are within 30%; for this reason, with a very good approximation, $\delta M/M$ is constant and we can assume that RMS(M)/<M> is the same for the three unknown mass distributions (hereafter also named *templates*). The accuracy of this approximation is even better for Li isotopes and B isotopes due to the small mass difference.

### 2.1. Template Transformations

We can define $T_7$, $T_9$ and $T_{10}$ as the unknown normalized templates for $^7$Be, $^9$Be and $^{10}$Be respectively.

A template $T_a$ can transform in the template $T_b$ by applying the operator $A_{a,b}T_a(x) = T_b(x)$, and we can assume $A_{a,b}$ is just transforming the coordinates $x \to g(x)$; therefore, to ensure template normalization:

$$T_b(x) = A_{a,b}T_a(x) = \frac{dg}{dx}T_a(g(x)) \tag{2}$$

In principle an infinite set of functions $g(x)$ is able to perform a transformation between two specific templates; however, we are typically interested in monotonic functions preserving quantiles while avoiding folding the template. A very simple set of transformations are the linear ones $L_{a,b}$ defined by combinations of translation and scale transformations: $x \to mx + q$.

The linear $L_{a,b}$ transforms a normal distribution in a normal distribution.

By defining $\sigma_a$ as the RMS of template $T_a$, and $x_a$ the median of template $T_a$, the linear transformation $L_{a,b}T_a = T_b$ is the function: $x \to \frac{\sigma_a}{\sigma_b}x + \left[x_a - \frac{\sigma_a}{\sigma_b}x_b\right]$.

The same transformation applied to a different template $L_{a,b}T_c = T_d$ provides: $\sigma_d = \sigma_c\frac{\sigma_b}{\sigma_a}$ and $x_d = x_b + (x_c - x_a)\frac{\sigma_b}{\sigma_a}$.

The linear transformation that satisfies the assumption, $\delta M/M = $ constant, is simply: $x \to \frac{x_a}{x_b}x$; that is a pure scaling, thereby depending only on the known beryllium isotope mass ratios and not on the unknown mass resolution or template shapes. In the following, the linear approximation for the template transformation is adopted.

### 2.2. Data-Driven Template Evaluation

Defining the known (measured) data distribution $D(x)$ and assuming as fixed the three fractions $^n$Be/Be, this equation system can be considered:

$$\begin{aligned} D(x) &= {}^7Be\ T_7 + {}^9Be\ T_9 + {}^{10}Be\ T_{10} \\ L_{7,9}D(x) &= {}^7Be\ T_9 + {}^9Be\ L_{7,9}T_9 + {}^{10}Be\ L_{7,9}T_{10} \\ L_{7,10}D(x) &= {}^7Be\ T_{10} + {}^9Be\ L_{7,10}T_9 + {}^{10}Be\ L_{7,10}T_{10}; \end{aligned} \tag{3}$$

therefore, the $^7$Be template can be written as:

$$T_7 = \frac{1}{^7Be}\left[D - \frac{^9Be}{^7Be}L_{7,9}D - \frac{^{10}Be}{^7Be}L_{7,10}D\right] + \tag{4}$$

$$+ \left(\frac{^9Be}{^7Be}\right)^2 T_{G1} + \frac{^9Be}{^7Be}\frac{^{10}Be}{^7Be}(T_{G2} + T_{G3}) + \left(\frac{^{10}Be}{^7Be}\right)^2 T_{G4}$$

and the last four terms, ghost-templates, are defined by:

$$\begin{aligned}
T_{G1} &= L_{7,9}T_9 = L_{7,x_{G1}}T_7 \\
T_{G2} &= L_{7,9}T_{10} = L_{7,x_{G2}}T_7 \\
T_{G3} &= L_{7,10}T_9 = L_{7,x_{G3}}T_7 \\
T_{G4} &= L_{7,10}T_{10} = L_{7,x_{G4}}T_7.
\end{aligned} \tag{5}$$

The median values of ghost-templates can be evaluated as follows:

$$\begin{aligned}
x_{G1} &= x_9 + (x_9 - x_7)\frac{\sigma_9}{\sigma_7} \simeq 11.5 \text{ amu} \\
x_{G2} &= x_9 + (x_{10} - x_7)\frac{\sigma_9}{\sigma_7} \simeq 13 \text{ amu} \\
x_{G3} &= x_{10} + (x_9 - x_7)\frac{\sigma_{10}}{\sigma_7} \simeq 13 \text{ amu} \\
x_{G4} &= x_{10} + (x_{10} - x_7)\frac{\sigma_{10}}{\sigma_7} \simeq 14 \text{ amu}
\end{aligned} \tag{6}$$

Profiting from the fact that the ghost-templates are placed beyond $T_{10}$ and that we know $^7$Be $> {}^9$Be $> {}^{10}$Be, the contribution of ghost-templates to Equation (4) is small and $T_7$ can be iteratively evaluated from measured data by using Equation (5).

Once $T_7$ is obtained, the other templates can be straightforwardly obtained by using $L_{7,9}$ and $L_{7,10}$, and a $\chi^2$ value for the fixed $^7$Be/Be and $^9$Be/$^{10}$Be can be obtained by comparison of the sum of the three weighted templates with $D(x)$. The $^7$Be/Be and $^9$Be/$^{10}$Be best fit configuration is retrieved by minimizing the $\chi^2$ value.

In a very similar way, the data-driven approach can be adopted for the measurements of a $^{11}$B/$^{10}$B or $^7$Li/$^6$Li flux ratio in cosmic rays. When applied to the measurement of the abundance ratio among only two isotopes, the data-driven approach formulation is even simpler with respect to the three Be isotopes; however, the missing $^8$Be, and the fact that $^7$Be $> {}^9$Be $> {}^{10}$Be, are two favorable conditions for the determination of the isotope templates directly from experimental data.

In particular, for the example of Li isotopes, the apparently simple solution for the $T_6$ template:

$$T_6 = \frac{1}{^6Li}\left[D - \frac{^7Li}{^6Li}L_{6,7}D\right] + \left(\frac{^7Li}{^6Li}\right)^2 L_{6,7}T_7 \tag{7}$$

would provide an inaccurate determination of the rightmost $T_6$ tail due to $^7$Li/$^6$Li $\sim 1$. This problem can be overcome by evaluating the rightmost tail by iteratively solving the $T_7$ template (that conversely would provide an inaccurate determination of its leftmost tail for the same reason):

$$T_7 = \frac{1}{^7Li}\left[D - \frac{^6Li}{^7Li}L_{7,6}D\right] + \left(\frac{^6Li}{^7Li}\right)^2 L_{7,6}T_6. \tag{8}$$

This data-driven approach has been tested on the Monte Carlo simulated events for lithium and beryllium isotopes in the AMS-02 spectrometer (see e.g., [10]), and it is able to correctly retrieve the injected isotopic ratios and template shapes within the statistical fluctuations.

### 3. An Example of an Application to Pamela Lithium and Beryllium Events

The application of the data-driven approach to lithium and beryllium events published by PAMELA experiment [9] is shown in the following as an example. It is important to note that despite a preliminary measurement of $^{10}Be/^9Be$ being obtained in the 0.25–0.85 GeV/n kinetic energy range, a full data analysis using this approach by the PAMELA-collaboration could provide a complete measurement of $^{10}Be/^9Be$ at up to $\sim$1 GeV/n.

In Figure 2, the example of lithium isotope measurements, by analyzing PAMELA-ToF events in the 0.25–0.75 GeV/n region, is shown. The templates for $^6Li$ and $^7Li$ were obtained by applying the data-driven approach.

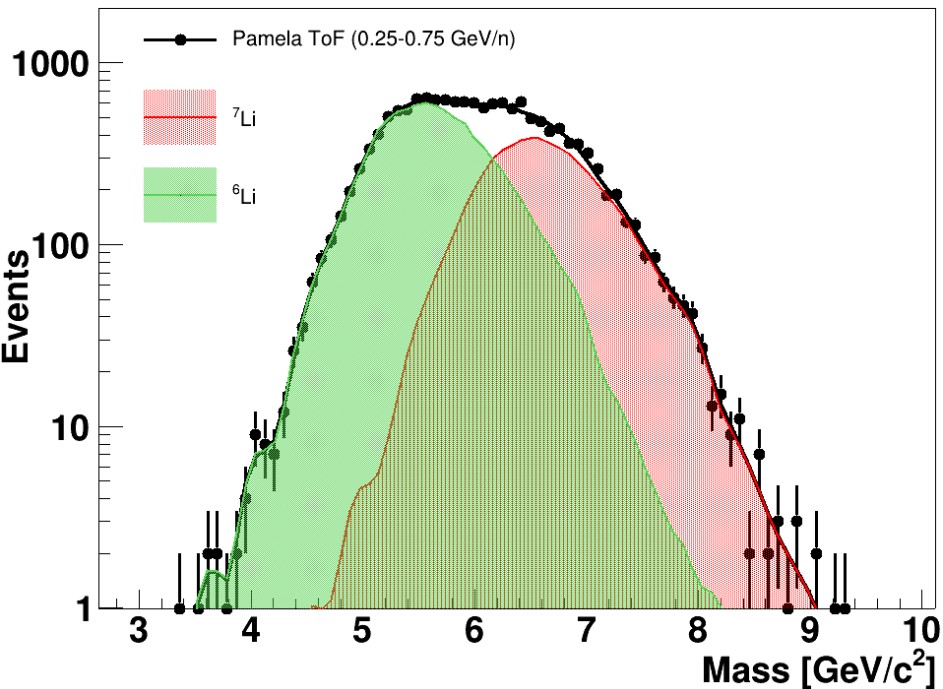

**Figure 2.** An example of lithium isotope measurements with the data-driven analysis of PAMELA-ToF data collected in the 0.25–0.75 GeV/n range.

It is important to note that trivial solutions of the data-driven analysis (Equations (7) and (8)) are obviously $^6Li/Li = 1$ and $^7Li/Li = 1$; these naive solutions are characterized by $\chi^2 = 0$. Therefore, the evaluation of the confidence interval for the local minimum of $\chi^2$ determined by the physical solution requires some care; in particular, the statistical bootstrap [11] was adopted to safely evaluate the confidence intervals. Figure 3 shows the $\chi^2$ minimum corresponding to the physical solution for the measurement of lithium isotope abundances using PAMELA ToF in the range 0.25–0.75 GeV/n. Figure 3 also shows the probability distribution of best-fit configurations obtained by statistically bootstrap-re-sampling the measured data distribution.

A similar approach was adopted for the data-driven analysis of beryllium isotopes; and in this case $^7Be/Be = 1$, $^9Be/Be = 1$ and $^{10}Be/Be = 1$ are three trivial solutions of Equation (4). In Figure 4, the $\chi^2$ map for the $<^7Be/Be$ vs. $^{10}Be/^9Be>$ parameter space is shown as an example of PAMELA-ToF events in the 0.65–0.85 GeV/n range (the last bin of this beryllium analysis).

In this last case the shown confidence interval was determined as the iso-$\chi^2$ contour containing 68% of the best-fit configurations obtained by statistically bootstrap-re-sampling the measured data distribution.

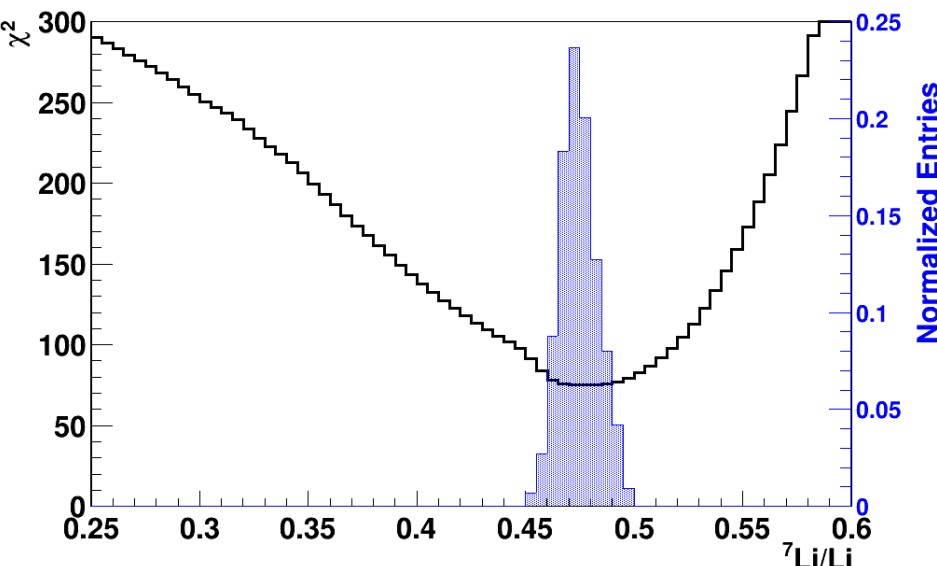

**Figure 3.** $\chi^2$ configurations obtained by data-driven analysis for lithium events collected by PAMELA-ToF in the 0.25–0.75 GeV/n region (black line). The best-fit probability distribution obtained by statistical bootstrap is shown for comparison (blue histogram).

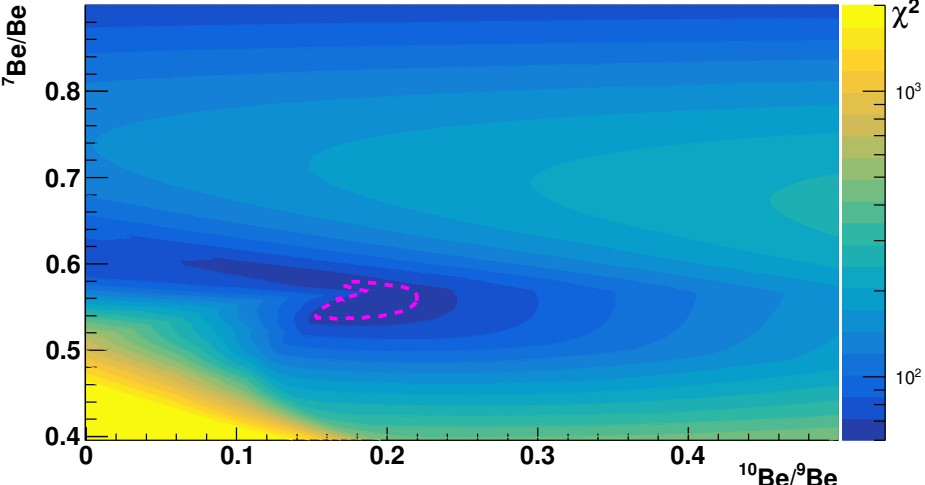

**Figure 4.** A map of $\chi^2$ configurations for beryllium events collected by PAMELA-ToF in the 0.65–0.85 GeV/n region. The 68% confidence interval is shown as a red dashed contour.

In Figure 5, the best-fit for the example of beryllium isotopes collected by PAMELA-ToF in the 0.65–0.85 GeV/n region is shown. The mass templates were obtained with the data-driven approach. Figure 5 also shows the same data and templates but as a function of $|M - 10|$; this visualization has the virtue of improved clarity for the $^{10}$Be evidence.

Finally, it is important to note that the results of this data-driven approach are identical, by construction, even applying an arbitrary/overall scaling of the mass values. For this reason, the results obtained by data-driven analysis are quite solid regarding possible rigidity/velocity scale miscalibrations that could prevent the traditional MC-based analysis, as shown in [9]. As a practical example, we applied the data-driven analysis to events measured by PAMELA calorimeter (Figure 4 of [9]), and without a tuned Monte Carlo model/calibration for the $dE/dx$ measurement, no less.

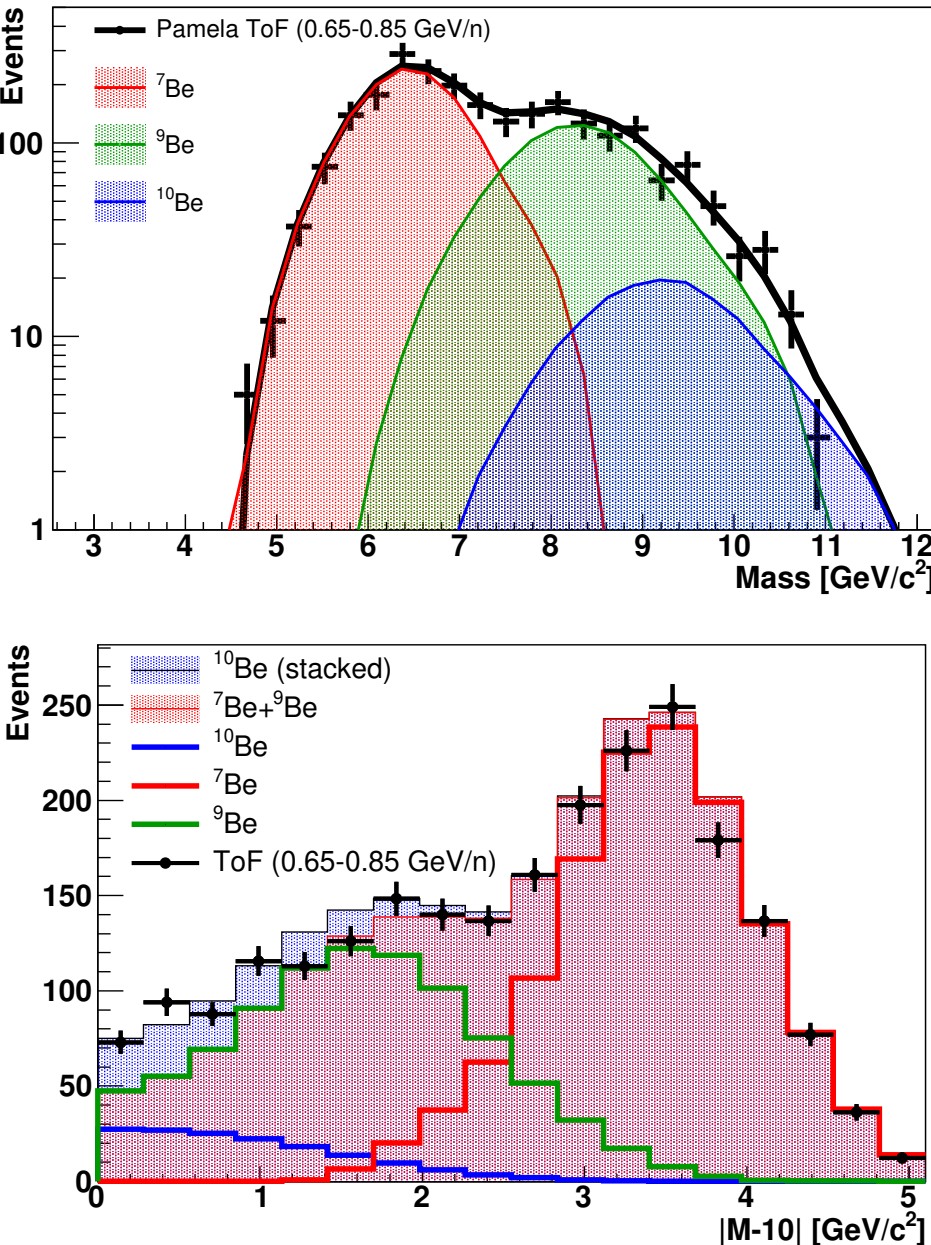

**Figure 5.** An example of beryllium isotope measurements with the data-driven analysis of PAMELA-ToF data collected in the 0.65–0.85 GeV/n range. The black continuous line is the sum of the three beryllium components. In the bottom plot the same data are shown as a function of $|M - 10|$l this allows one to stack the $^{10}$Be contribution (blue filled) over the $^7$Be + $^9$Be (red filled).

## 4. Results and Discussion

The results of the data-driven analysis of $^7$Li/$^6$Li, $^7$Be/Be and $^{10}$Be/$^9$Be ratios applied to PAMELA data [9] are reported in Table 1 and shown in Figures 6–8 along with previous experiment measurements [12–24].

**Table 1.** Results of the data-driven analysis applied to PAMELA Li and Be events.

| $E_k/n$ [GeV/n] | $^7Li/\,^6Li$ | |
|---|---|---|
| 0.15–0.35 (Calo) | $0.95 \pm 0.03 \pm 0.2$ | |
| 0.35–0.75 (Calo) | $0.94 \pm 0.03 \pm 0.25$ | |
| 0.15–0.35 (ToF) | $0.96 \pm 0.03 \pm 0.2$ | |
| 0.35–0.75 (ToF) | $0.88 \pm 0.04 \pm 0.25$ | |
| $E_k/n$ [GeV/n] | $^7Be/Be$ | $^{10}Be/\,^9Be$ |
| 0.2–0.52 (Calo) | $0.56 \pm 0.01 \pm 0.03$ | $0.12 \pm 0.02 \pm 0.07$ |
| 0.25–0.45 (ToF) | $0.53 \pm 0.01 \pm 0.03$ | $0.115 \pm 0.01 \pm 0.07$ |
| 0.45–0.65 (ToF) | $0.56 \pm 0.01 \pm 0.035$ | $0.15 \pm 0.02 \pm 0.07$ |
| 0.65–0.85 (ToF) | $0.56 \pm 0.02 \pm 0.04$ | $0.17 \pm 0.03 \pm 0.07$ |

The data-driven measurements obtained by analyzing PAMELA-ToF events (black dots) are in reasonable agreement with the measurements obtained with the PAMELA calorimeter (blue square); and regarding $^7Li/^6Li$ and $^7Be/Be$, the results of the data-driven analysis are in agreement with the ones published in [9] based on the Monte Carlo template fit of the PAMELA data (orange dots). The green shaded regions in Figures 6–8 are conservative estimations of the systematic errors for the data-driven analysis, related to the possible departures from the assumption of pure template scaling, thereby considering $\delta M/M = K\,(1 \pm \alpha_M)$ where K is a constant and $\alpha_M$ is a possible, small isotope dependent correction. In particular, knowing the measured PAMELA rigidity resolution [25] in the considered 1.5–4 GV range, and knowing that, for a fixed velocity, the rigidity of $^7Be$ is 70% of the rigidity of $^{10}Be$, a conservative upper limit of $\alpha_M < 10\%$ can be inferred for the possible departures from the exact template scaling relation. A complete evaluation of systematic uncertainties requires a study of the possible differences in the selection acceptance for $^7Be, ^9Be$ and $^{10}Be$ as well, and of the different types of contamination due to B, C, N and O fragmentation crossing the material above the detector. These systematics cannot be estimated without a Monte Carlo simulation of the detector; however, their contributions are expected to be small (a few percent) with respect to the wide uncertainties plotted in Figures 7 and 8. Similar arguments also hold for the $^7Li/^6Li$ flux ratio measurements.

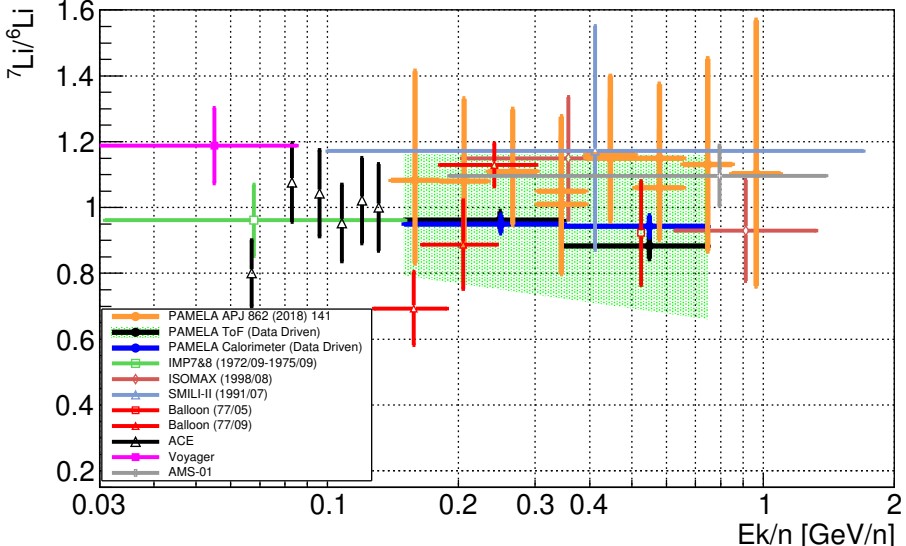

**Figure 6.** Results of the data-driven measurement of the $^7Li/^6Li$ ratio compared with previous experimental results and with a Monte Carlo based analysis of PAMELA [9] (orange dots). The green shaded contour is the systematic uncertainty inferred for the data-driven analysis.

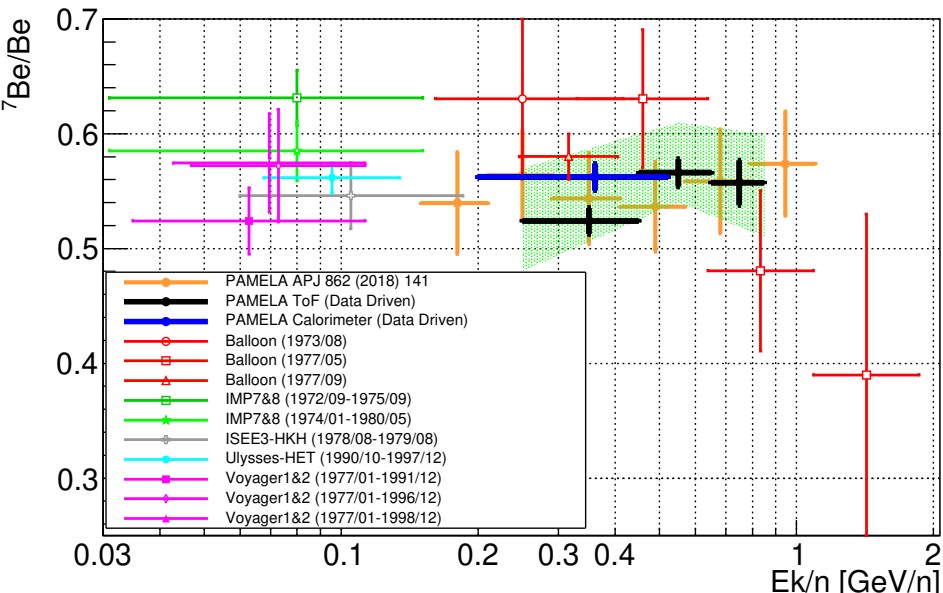

**Figure 7.** Results of the data-driven measurement of the $^7$Be/Be fraction compared with previous experimental results and with a Monte Carlo based analysis of PAMELA [9] (orange dots). The green shaded contour is the systematic uncertainty inferred for the data-driven analysis.

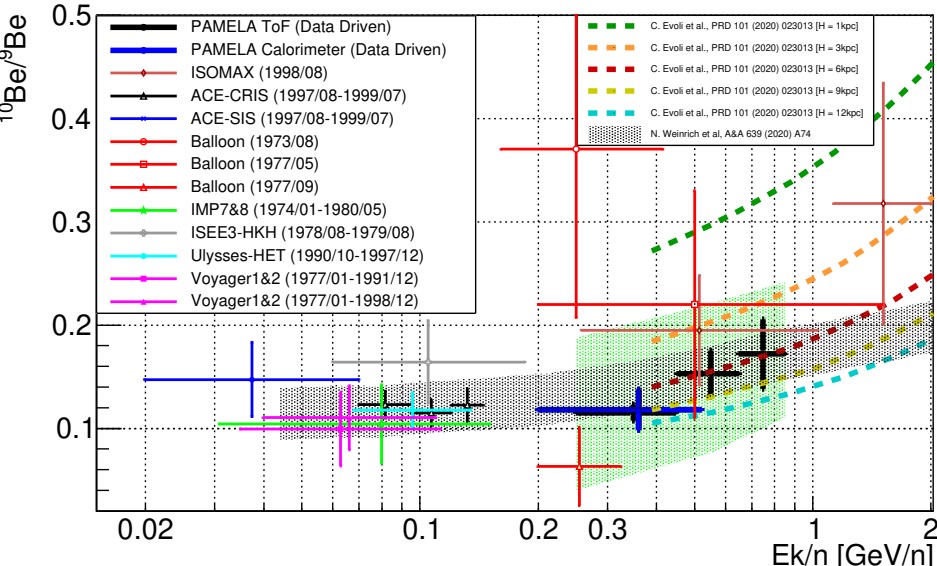

**Figure 8.** Results of the data-driven measurement of the $^{10}$Be/$^9$Be ratio compared with previous experimental results and theory expectations: [1] (dashed lines) and [4] (hatched region). The green shaded contour is the systematic uncertainty inferred for the data-driven analysis.

The new information provided by the data-driven analysis, when applied to PAMELA data, is a relatively precise estimation of $^{10}$Be/$^9$Be ratio in the range 0.2–0.85 GeV/n, where existing measurements are scarce and affected by large uncertainties. In particular, it is interesting to note that these measurements strengthen the previous indications for a rising $^{10}$Be/$^9$Be ratio at high kinetic energy and are in good agreement with the models of [1,2,4,5], which provided predictions of $^{10}$Be/$^9$Be tuned with the up-to-date AMS-02 fluxes (and previous $^{10}$Be/$^9$Be measurements).

In Figure 8, the comparison of the $^{10}$Be/$^9$Be flux ratio with the expectations for different values of the *H* parameter in the model [1] (dashed lines) confirms the current knowledge for this parameter in the range 3–8 kpc.

To further study the capability of current measurements of $^{10}$Be/$^9$Be to act as a radioactive clock providing information about cosmic ray propagation time in the galaxy, we

plotted in Figure 9 the existing $^{10}Be/^9Be$ data as a function of the inverse of the relativistic Lorentz factor, $\gamma$. This representation allows a simple and minimal phenomenological model to quantify the energy dependence of the $^{10}Be/^9Be$ ratio:

$$\frac{^{10}Be}{^9Be} = Ae^{-\frac{T}{\gamma\tau}} \tag{9}$$

where $\tau = 2$ My is the lifetime of $^{10}Be$ radioactive decay at rest, and the two parameters $A$ and $T$ can be viewed, respectively, as the average $^{10}Be/^9Be$ ratio produced by the primary cosmic ray collisions and the average propagation time. It is important to remember that Equation (9) is a crude but simple model; other effects not related to cosmic ray propagation time could contribute to the energy dependence of $^{10}Be/^9Be$ as—for example, the existence of an underdense bubble in the local interstellar medium [26].

The fit of the existing measurements with the simple model of Equation (9) is drawn as a dashed line in Figure 9 and provides A = 0.27 $\pm$ 0.13 and T = 1.9 $\pm$ 1.1 My. Removing the PAMELA data from this fit would increase the uncertainty of T by 30%; moreover, the uncertainty of A would be more than double.

In conclusion, PAMELA information on the $^{10}Be/^9Be$ ratio provided by the data-driven approach is important for the study of cosmic ray propagation. Future precision measurements are expected from the forthcoming results of AMS-02 and HELIX experiments.

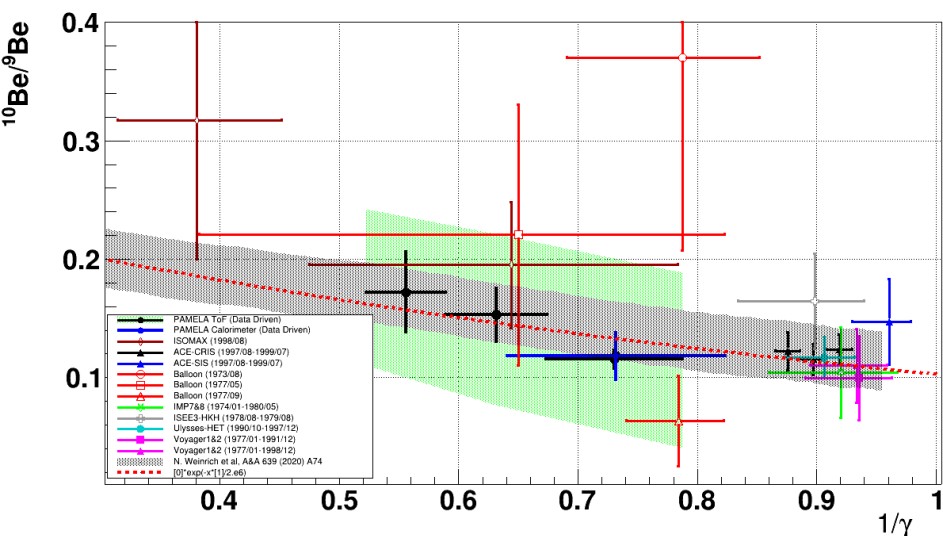

**Figure 9.** Measurements of the $^{10}Be/^9Be$ flux ratio as a function of $1/\gamma$. The dashed line is a phenomenological exponential fit. The new measurements obtained by the Pamela experiment using this data-driven approach strengthen the experimental indications of the expected $1/\gamma$ behavior.

**Author Contributions:** Conceptualization, F.N.; methodology, F.N.; software, F.N. and C.C.; validation, F.N. and C.C.; formal analysis, F.N.; investigation, F.N.; data curation, C.C.; writing—original draft preparation, F.N.; writing—review and editing, F.N. and C.C.; visualization, F.N. and C.C.; supervision, F.N.; All authors have read and agreed to the published version of the manuscript.

**Funding:** This research received no external funding.

**Institutional Review Board Statement:** Not applicable.

**Informed Consent Statement:** Not applicable.

**Conflicts of Interest:** The authors declare no conflict of interest.

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
