# Peer review of "Beryllium Radioactive Isotopes as a Probe to Measure the Residence Time of Cosmic Rays in the Galaxy and Halo Thickness: A “Data-Driven” Approach†"

_universe, doi:10.3390/universe7060183_

Round 1
Reviewer 1 Report
This is a very useful addition to the measurement of the important 10Be/9Be ratio, especially since it addresses energies where the Lorentz factor increases the 10Be decay time to a range which is more constraining for the cosmic-ray containment time. This was previously only done by the ISOMAX experiment which unfortunately failed on a later flight.
I consider the manuscript acceptable as is, but suggest the new experimental values given in the Figures, especially Fig 9, should be given in a table or in the text so that readers can make use of the data in their modelling of cosmic-ray propagation.
typo: line 191 kPc -> kpc
Author Response
We thank Reviewer1 for reviewing the manuscript.
We have modified the manuscript according to Reviewer1 suggestions.
Suggestion1)
I consider the manuscript acceptable as is, but suggest the new experimental values given in the Figures, especially Fig 9, should be given in a table or in the text so that readers can make use of the data in their modelling of cosmic-ray propagation.
Answer1)
We have added a table (Table1) containing the experimental values obtained in this Data Driven analysis for the Pamela Li and Be events.
Suggestion2) typo: line 191 kPc -> kpc
Answer2) The typo has been corrected (thanks)
Best regards
Reviewer 2 Report
The submitted manuscript presents an interesting re-analysis of PAMELA data on Beryllium and Lithium cosmic rays aiming at determining the relative abundances of their isotopes. In particular, determining the relative abundance of Be9 and Be10 is very important to remove the degeneracy between the Halo thickness and the CR diffusion coefficient.
The manuscript is well written and clear. The scientific topic is relevant for the community of High Energy Astrophysics and the results are significant and well presented.
I can recommend the manuscript for publication.
I have only a couple of minor comments.
1) In page 6, after Eq.(8) it is written that the same "Data-Driven approach has been tested on the Monte Carlo simulated events for Li and Be isotopes in the AMS-02 spectrometer" than the Authors cite reference [10], however such a reference discuss the Helium flux and their isotopes, hence this sentence is not clear to me.
2) In figure 8, the hatched region reporting the theoretical model by Weinrich et al.(2020) is almost invisible and difficult to see. I suggest to make it slightly darker.
3) Galaxy and Galactic should be written with capital "G" when refer to the Milky Way.
Author Response
We thank Reviewer2 to review this manuscript.
Here the answer to Reviewer2 questions:
Question1)
1) In page 6, after Eq.(8) it is written that the same "Data-Driven approach has been tested on the Monte Carlo simulated events for Li and Be isotopes in the AMS-02 spectrometer" than the Authors cite reference [10], however such a reference discuss the Helium flux and their isotopes, hence this sentence is not clear to me.
Answer1)
Among the many references to AMS-02 We choose to cite the one that is currently the unique isotopic measurement of AMS-02.
We modified the sentence adding: (see e.g. [10])
Question2)
In figure 8, the hatched region reporting the theoretical model by Weinrich et al.(2020) is almost invisible and difficult to see. I suggest to make it slightly darker.
Answer2)
We modified the figure 8 (and also the figure 9) using a darker hatching for the theoretical model by Weinrich et al.(2020)
Question3) Galaxy and Galactic should be written with capital "G" when refer to the Milky Way.
Answer3) We correct all the occurrences of Galaxy and Galactic.
Thank you
Reviewer 3 Report
In this manuscript, the authors present an original data driven method to extract measurement of the 10Be/9Be isotopic ratio in cosmic rays.
The results are of considerable interest and timeliness. The methodology appears solid. The manuscript is well written and structured.
In my opinion, this manuscript is suitable for publication in Universe. I have only a series of minor comments and suggestions I would like the authors to address. I do not need to see the manuscript again.
1) Abstract: please correct: time dilatation->dilation.
2) Introduction: the authors should at least mention the possibility of extracting the CR diffusion parameters H/D using the elemental ratio Be/B in place of the isotopic ratio 10Be/9B. This possibility is discussed Webber & Soutol ApJ 2010 and, more recently, it is analyzed in Tomassetti arXiv:1509.05776 and Evoli et al. arXiv:1910.04113.
3) Introduction: the problem of uncertainties or biases in fragmentation cross-sections should also be mentioned. In fact, the possibility of breaking the H/D degeneracy using isotopic Be data relies heavily on the knowledge of their production cross sections.
4) Results: the authors try to quantify the energy dependence of the measured 10Be/9Be ratio with a simple 2-parameter model given in Eq. (9). However it should be mentioned that a prediction for the gamma dependence of the ratio can be obtained with CR propagation models. Moreover, such a dependence can be appreciably affected by inhomogeneous gas distribution in the local ISM due to the presence of the Local Bubble. In fact, secondary 10Be isotopes are produced by fragmentation of CRs with interstellar matter, but in the region within 100-200 kpc from Earth the gas density is depleted. Such a local void, called Local Bubble, causes an appreciable impact on the energy dependence of the 10Be/9Be ratio, because unstable 10Be must originate from short distances. This effect is explained well in Donato et al. arXiv:0108079.
Author Response
We thank Reviewer2 for reviewing the manuscript and for very useful suggestions.
We have modified the manuscript based on the comments of the Reviewer2.
Q1) Abstract: please correct: time dilatation->dilation.
A1) Typo corrected (thanks)
Q2) Introduction: the authors should at least mention the possibility of extracting the CR diffusion parameters H/D using the elemental ratio Be/B in place of the isotopic ratio 10Be/9B. This possibility is discussed Webber & Soutol ApJ 2010 and, more recently, it is analyzed in Tomassetti arXiv:1509.05776 and Evoli et al. arXiv:1910.04113.
Q3) Introduction: the problem of uncertainties or biases in fragmentation cross-sections should also be mentioned. In fact, the possibility of breaking the H/D degeneracy using isotopic Be data relies heavily on the knowledge of their production cross sections.
Answer to Q2 and Q3)
We improve the introduction adding lines 47-56 to mention both points as suggested by the Reviewer2. Also Reference [6]-[7]-[8] has been added.
Q4) Results: the authors try to quantify the energy dependence of the measured 10Be/9Be ratio with a simple 2-parameter model given in Eq. (9). However it should be mentioned that a prediction for the gamma dependence of the ratio can be obtained with CR propagation models. Moreover, such a dependence can be appreciably affected by inhomogeneous gas distribution in the local ISM due to the presence of the Local Bubble. In fact, secondary 10Be isotopes are produced by fragmentation of CRs with interstellar matter, but in the region within 100-200 kpc from Earth the gas density is depleted. Such a local void, called Local Bubble, causes an appreciable impact on the energy dependence of the 10Be/9Be ratio, because unstable 10Be must originate from short distances. This effect is explained well in Donato et al. arXiv:0108079.
Answer 4)
We improved the "Results and discussion" section following the Reviewer2 suggestion, thus adding a sentence (line 206-208) mentioning the possible effect of the local bubble. Also the reference [26] has been added.
Best regards